# Prime editor with rational design and AI-driven optimization for reverse editing window and enhanced fidelity

Chao Yang [1,2,3,4,5,15] ✉, Qingxiao Fang[1,2,3,4,5,15], Mengyu Li[1,2,3,4,5,15], Jin Zhang[1,2,3,4,5,15], Rui Li[6], Tianxing Zhou[1,2,3,4,5], Keshan Wang [7], Jie Deng [8], Xiuchao Wang[1,2,3,4,5], Chongbiao Huang[1,2,3,4,5], Yukuan Feng[1,2,3,4,5], Xiaoping Zhang[7], Lei Shi [9,10,11,12,13] ✉, Changhao Bi [14] ✉, Xueli Zhang [14] ✉, Jun Yu [1,2,3,4,5] ✉ & Jihui Hao [1,2,3,4,5] ✉

Prime editing (PE) is a precise tool for introducing genetic mutations in eukaryotes. Extending the efficient editing scope and mitigating undesired byproducts are possible. We introduce reverse PE (rPE), a SpCas9-directed variant that enabled DNA editing at the 3' direction of HNH-mediated nick site. The rPE leveraging nCas9-D10A and rPE gRNA targeting the 5' direction of HNH-mediated nick site inscribes genetic alterations, achieving a reverse editing window and potentially high fidelity. HNH and reverse transcriptase engineered using protein language models in conjunction with La facilitate circular erPEmax and erPE7max, achieving editing efficiency up to 44.41% without nick gRNA or positive selection. Furthermore, our findings underscore the capability of rPE in inserting functionally enhanced variant (*PIK3CD*[E527G]) for cell therapy. By expanding the editing scope and enhancing genomic manipulability, rPE represents a meaningful advancement in prime editing, improving its utility for research and therapeutic applications.

Prime editing (PE) is a precise method enabling targeted DNA modifications in eukaryotes without inducing double-strand breaks (DSBs). PE enables correction of a wide array of known pathogenic mutations[1–3]. PE comprises a programmable nickase, *Streptococcus pyogenes* Cas9 (SpCas9-H840A), reverse transcriptase (RT; typically, the Moloney Murine Leukemia Virus (MMLV) RT variant), and a PE guide RNA (pegRNA). The pegRNA encompasses a primer binding site (PBS), RT template (RTT), and spacer sequence. The Cas9-H840A binds to the pegRNA, forming a ribonucleoprotein (RNP) complex that specifies the target site in the genome. RuvC nuclease nicks the non-targeted strand (NTS, not paired with guide RNA) to generate a single-strand DNA primer for PBS binding. Subsequently, the desired edit in RTT is encoded through reverse transcriptional reaction and incorporated into the genome. The orchestrated design and multiple

[1]Pancreas Center, Tianjin Medical University Cancer Institute and Hospital, Tianjin, China. [2]National Clinical Research Center for Cancer, Tianjin, China. [3]State Key Laboratory of Druggability Evaluation and Systematic Translational Medicine, Tianjin, China. [4]Tianjin Key Laboratory of Digestive Cancer, Tianjin, China. [5]Tianjin's Clinical Research Center for Cancer, Tianjin, China. [6]College of Biotechnology, Tianjin University of Science and Technology, Tianjin, China. [7]Department of Urology, Union Hospital, Tongji Medical College, Huazhong University of Science and Technology, Wuhan, China. [8]School of Chemistry and Chemical Engineering, Huazhong University of Science and Technology, Wuhan, China. [9]State Key Laboratory of Experimental Hematology, Tianjin, China. [10]Key Laboratory of Breast Cancer Prevention and Therapy (Ministry of Education), Tianjin, China. [11]Key Laboratory of Immune Microenvironment and Disease (Ministry of Education), Tianjin, China. [12]The Province and Ministry Co-sponsored Collaborative Innovation Center for Medical Epigenetics, Tianjin, China. [13]Department of Biochemistry and Molecular Biology, School of Basic Medical Sciences, Tianjin Medical University, Tianjin, China. [14]Tianjin Institute of Industrial Biotechnology, Chinese Academy of Sciences, Tianjin, China. [15]These authors contributed equally: Chao Yang, Qingxiao Fang, Mengyu Li, Jin Zhang. ✉ e-mail: yangchao@tjmuch.com; shilei@tmu.edu.cn; bi_ch@tib.cas.cn; zhang_xl@tib.cas.cn; yujun@tjmuch.com; haojihui@tjmuch.com

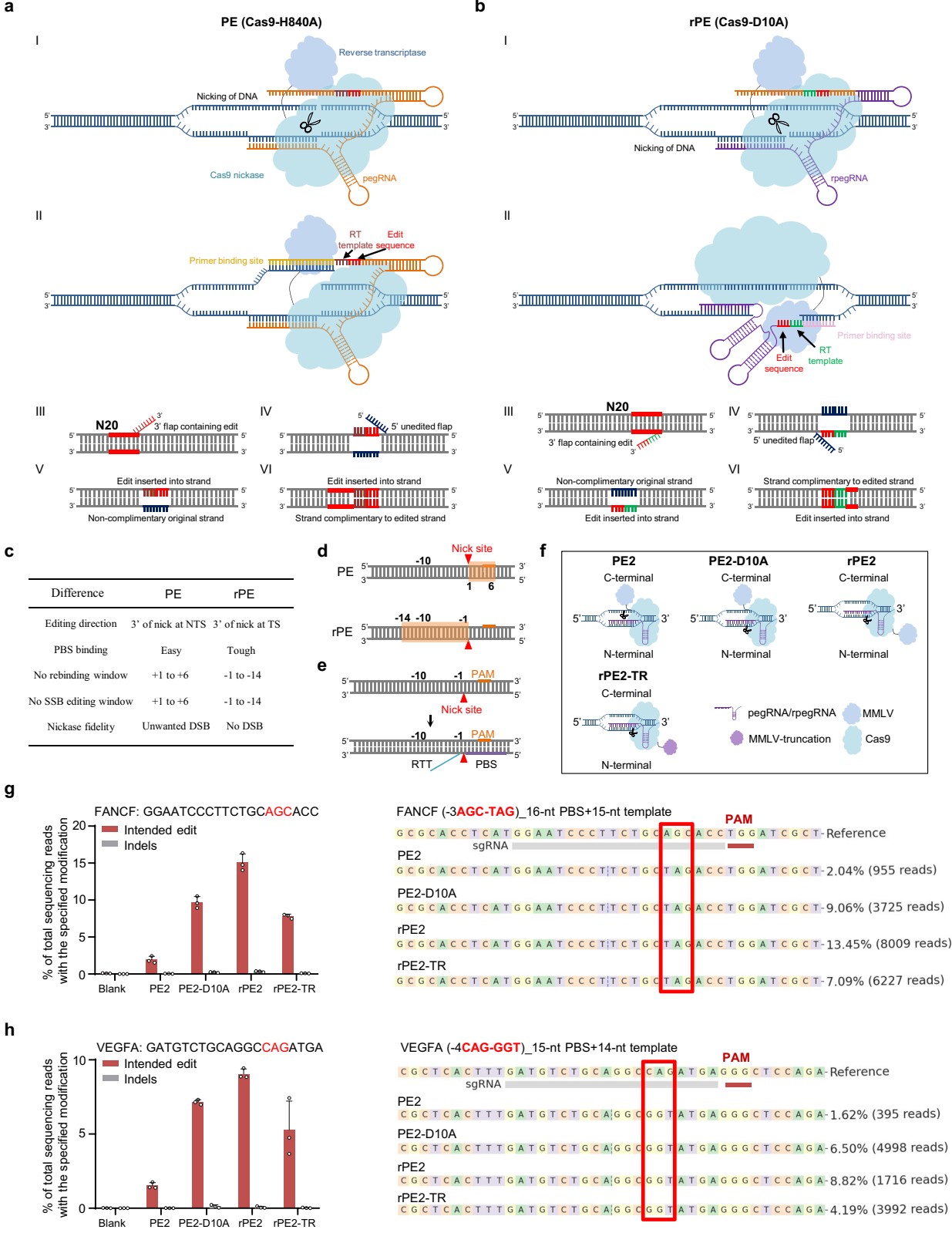

components of PE have led to the development of versatile genome editing tools with significant potential in genetic therapy.

Efforts have been directed towards enhancing the editing efficiency and expanding the application scenarios of PE. Mismatch repair (MMR) signaling has been identified as a factor that negatively affects PE editing, with the potent inhibitor MLH1dn for MMR utilized to improve PE editing efficiency[4]. Recently, RNA-binding protein and

epigenetic strategies have also been utilized to improve the editing function[5,6]. Additionally, susceptibility to nuclease degradation at the 3' of pegRNA in eukaryotes has been addressed by incorporating structured RNA motifs or generating circular RNA in split PE[7,8]. The split strategy has also been employed in developing highly efficient Cas12-directed PE variants[9]. Evolved Cas9 and compact RT variants also reportedly enhance PE editing, further increasing its therapeutic

**Fig. 1 | Design and construction of the rPE. a** Overview of canonical prime editing. Steps I–VI outline the sequence of events for genetic alteration via canonical PE. The black arrow in step II indicates the editing direction, while the red line in steps III–VI represents the spacer sequence. **b** Overview and design of rPE. Steps I–VI illustrate the sequence of events for genetic alteration via rPE. The black arrow in step II shows the editing direction, with the red line in steps III–VI representing the spacer sequence. **c** Table comparing PE and rPE for editing direction, PBS binding, no rebinding window, no continuous SSB editing window, and nickase fidelity. NTS non-targeted strand, TS targeted strand. **d** Schematic depicting the potentially no rebinding and continuous SSB editing window in PE versus rPE. Red arrow indicates

the nick site, and the orange line represents the PAM sequence. Editing position 0 corresponds to the nick site. **e** Schematic of the rpegRNA and its components. The purple line represents the PBS binding to the 5′ direction of the nick site, while RTT is incorporated in the 3′ direction of the nick site. **f** Schematic of PE2, PE2-D10A, rPE2, and rPE2-TR (truncation). Comparison of editing efficiency (red bars) and indel frequency (gray bars) at FANCF (**g**) and VEGFA (**h**) loci across PE2, PE2-D10A, rPE2, and rPE2-TR in HEK293T cells (left). Genotype distribution and editing frequency at FANCF (**g**) and VEGFA (**h**) loci in HEK293T cells (right). Data are presented as mean ± SD from *n* = 3 independent biological replicates. Source data are provided as a Source Data file.

potential[10]. Strategies utilizing twin pegRNAs to edit both DNA strands have been developed, enabling >100 bp insertions and deletions (indels), thus broadening PE applications, including twinPE, PRIME-Del, and others[11–14]. Moreover, gene-sized indels (>5000 bp) have been achieved by combining serine recombinase and installing recombinase landing sites[15]. However, the editing scope of PE remains insufficiently explored.

The current editing scope of PE is limited to DNA editing at the 3′ direction of the RuvC-mediated nick site. Attempts to expand this scope include the utilization of *Francisella novicida* Cas9 (FnCas9) to nick at 6 bp upstream from the protospacer adjacent motif (PAM), and construction of PE-SpRY variant with PAM flexibility, albeit with reduced activity[16,17]. More importantly, optimal editing efficiency in PE is observed with limited lengths of PBS/RTT and restricted editing positions, as predicted from high-throughput analysis using deep learning[18]. In addition, when PE intended editing does not contain PAM or its adjacent 3 bp, the accumulation of single-strand breaks (SSBs) induced by RuvC nuclease raises concerns for secure gene therapy, potentially promoting genomic instability[19,20]. Accordingly, the efficient editing scope of canonical PEs could be further expanded.

In this study, we demonstrate the feasibility of reverse PE (rPE) and introduce versatile rPE systems capable of efficiently editing 3′ end sequences at HNH-mediated nick sites, achieving a reverse editing window with potentially high fidelity. By optimizing rpegRNA, HNH, and MMLV RT, we engineer rPE variants with improved editing capabilities and demonstrate its therapeutic potential in adoptive cell therapy.

## Results

### Design and construction of the rPE

Currently, despite the ability to target PAM and its adjacent three bases, the editing scope of PE is confined to the 3′ direction at the RuvC-mediated nick site (Fig. 1a). Notably, the editing efficiency decreases significantly with long RTTs, particularly hindering distal edits (at >= +12 position)[18], underscoring the necessity to broaden the efficient editing scope for PE. Furthermore, given that continuous SSBs could potentially be mitigated by editing the PAM or its adjacent spacer (+1 to +3) in canonical PE, shifting the editing window of PE to the 5′ direction at the RuvC-mediated nick site appears to fulfill the requirements of both extending the editing scope and eliminating of SSBs by directly editing the spacer sequence.

To achieve the transition of the editing window for PE, an inverse design for PBS and RTT at non-targeted strand was initially considered (Supplementary Fig. 1a), but it was discarded due to the catalytic limitations of RT. Subsequently, we endeavored to implement a rPE strategy at the targeted strand, thereby obtaining the opposite editing window (Fig. 1b). This transition of nicked DNA strand was attained by converting Cas9-H840A to Cas9-D10A, and the pegRNA was designed based on the targeted strand, with the PBS binding to the DNA sequence adjacent to the 5′ terminus of the HNH-mediated nick site (Fig. 1b). Notably, the majority of PBS binding bases are engaged as double-strand DNA in rPE, unlike the single-strand primer on the non-targeted strand in PE (Fig. 1c), potentially limiting the editing function. At the molecular level, an RNA-DNA hybrid forms during DNA repair or

transcription in eukaryotes[21]. Thus, the PBS, as single-strand RNA, might still form an RNA-DNA hybrid, facilitating reverse transcription in conjunction with RT. However, rPE provided a broader editing window without gRNA rebinding or continuous SSB production[22] (Fig. 1c, d). Furthermore, Cas9-H840A was reported to generate unwanted DSBs compared to Cas9-D10A[23], indicating a potentially higher fidelity for rPE (Fig. 1c).

To validate the rPE strategy, 3′-extended reverse pegRNAs (rpegRNA) were designed based on the DNA sequence of the targeted strand[2] (Fig. 1e). Several PE variants (Fig. 1f) were constructed following the reverse PE strategy, considering the modular structure of PE and differential activity of reverse transcriptase mutants, including PE2-D10A, rPE2, rPE2-TR (MMLV RT truncation). Subsequently, rpegRNAs targeting FANCF and VEGFA loci were transfected into HEK293T cells along with PE variants. Remarkably, substantial efficiency with a reverse editing window was observed with these variants, with rPE2 showing the highest efficiency of up to 16.34% (Fig. 1g, h). Additionally, the rPE strategy using saCas9 in HEK293T cells also yielded a reverse editing outcome (Supplementary Fig. 1b, c). Considering the significant impact of PBS and RTT lengths on PE editing efficiency[2], we then evaluated the effects of rpegRNAs with varying PBS and RTT lengths. The highest editing efficiency was achieved over a 4-day period with PBS and RTT lengths ranging from 10 to 16 nucleotides (nt), similar to the lengths commonly used in traditional PE (Supplementary Fig. 1d, e). To further assess the fidelity of rPE, byproducts generated by rPE2 and PE2 nickase were compared. Transfections of PE2 and rPE2 with the indicated gRNAs revealed a higher indel frequency in PE2 than in rPE2 at the FANCF-2 and CTLA loci (Supplementary Fig. 1f). The potentially continuous SSB production was also indirectly evaluated, and the data indicated a higher indel frequency when PE did not edit the PAM or its adjacent 3 bp compared to rPE (Supplementary Fig. 1g, h). Moreover, given the stringent binding requirement of PBS in rPE, genomic loci linked to varying levels of transcriptional activity were separately tested using rPE (Supplementary Fig. 1i). However, editing efficiency did not significantly differ across loci with distinct transcriptional activity (Supplementary Fig. 1j, k), suggesting a more complex underlying mechanism. Taken together, these findings demonstrated that rPE strategy implemented on the targeted strand induced substantial editing efficiency with a reverse editing window and potentially higher fidelity.

### Characterization of rPE in human cells

Based on the encouraging editing outcomes observed, we extended the evaluation of the rPE to additional genomic loci. The rPE2 and rPE2-TR were co-transfected with rpegRNAs designed to introduce single or multiple base mutations. Notably, the length of PBS and RTT was meticulously designed to range from 10 to 15 nt, with a G/C content of 40–60%, guided by previous findings[2]. Our results revealed that rPE2 and rPE2-TR precisely introduced mutations with an average editing efficiency ranging from 1.02 to 16.99% and 0.25 to 8.47%, respectively (Fig. 2a–c). The decreased editing efficiency observed with rPE2-TR might be due to the closer spatial proximity between the RT catalytic domain and Cas9. The efficacy of the rPE strategy was also confirmed at three genomic loci in HeLa and HepG2 cells (Supplementary Fig. 2a, b). To compare the editing capabilities

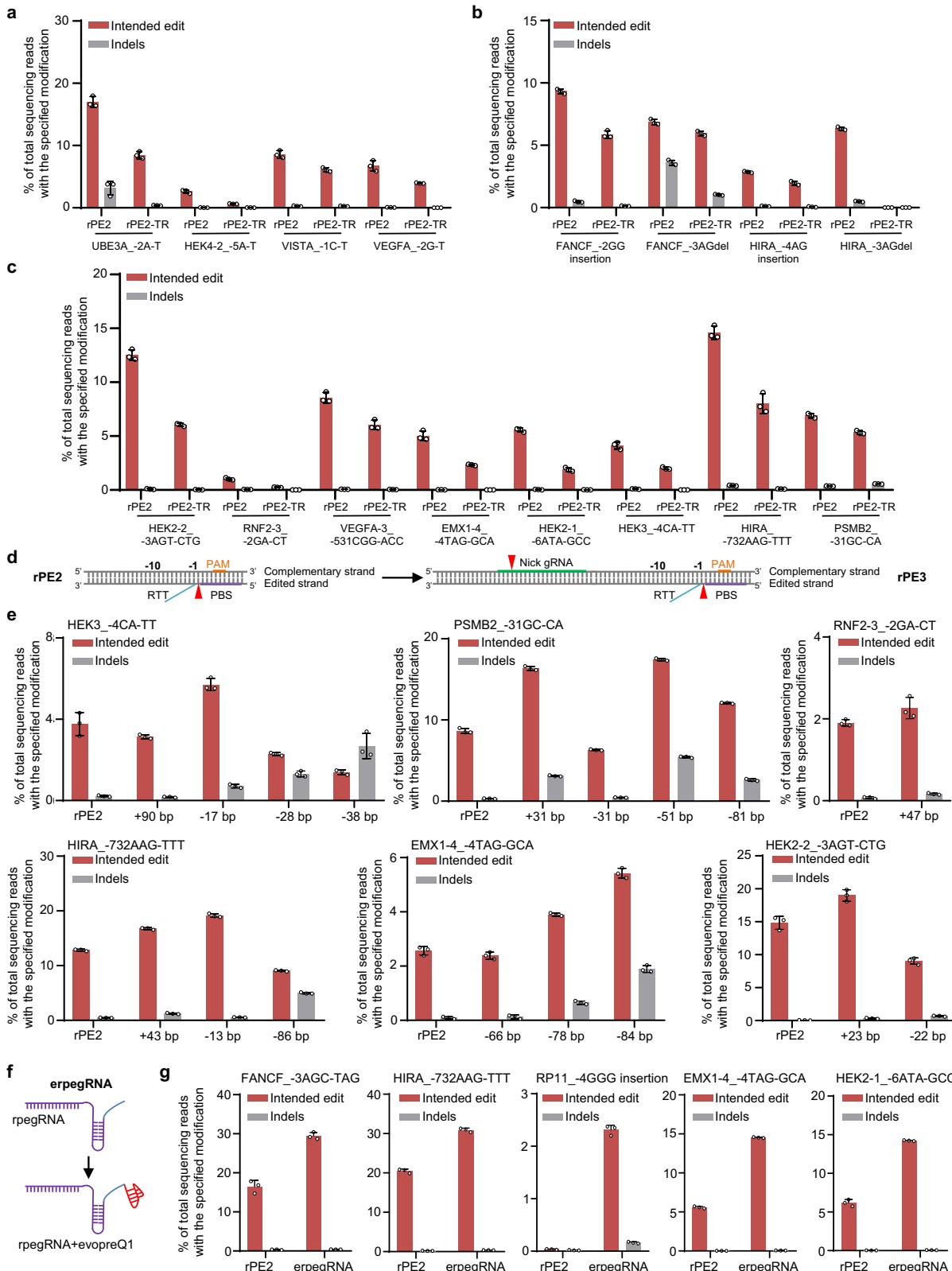

of rPE2 with canonical PE2, ten genomic loci were selected and targeted with the same spacer. Both systems successfully achieved the desired edits, but PE2 demonstrated a significantly higher editing efficiency (Supplementary Fig. 2c), hinting at the necessity for further improvement of rPE2.

To enhance the rPE efficiency, we introduced a nick gRNA (ngRNA)[2,24,25] to create the rPE3 (Fig. 2d). A significant increase in editing efficiency and indel frequency with rPE3, with EMX1-4 showing an increased efficiency of approximately 2.12-fold (Fig. 2e). Additionally, we also constructed rPE3b, where nicking of the non-edited strand occurs only after resolution of the edited strand flap (Supplementary Fig. 3a), aiming for lower indel byproducts. However, designing ngRNA based on the mutational DNA sequence in rPE3b posed a challenge as the desired complementary sequence did not form even after the

**Fig. 2 | Characterization of rPE in human cells.** Editing efficiency (red bars) and indel frequency (gray bars) of rPE2 and rPE2-TR at 16 genomic loci with single-base editing (**a**), short insertions or deletions (**b**), and multiple base substitutions (**c**) in HEK293T cells. Editing outcomes are indicated. Data are presented as mean ± SD from $n = 3$ independent biological replicates. **d** Schematic of the rPE3 strategy. The red arrow indicates the nick site, and the orange line represents the PAM sequence. The nick gRNA is marked with a green line. **e** Editing efficiency (red bars) and indel frequency (gray bars) of rPE2 and rPE3 with different ngRNAs at four genomic loci in HEK293T cells. The nick position is indicated. Data are presented as mean ± SD from $n = 3$ independent biological replicates. **f** Schematic of the erpegRNA strategy. The blue RNA hairpin represents the evopreQ1 sequence. **g** Comparison of editing efficiency (red bars) and indel frequency (gray bars) of rPE2 and rPE2 with erpegRNA across five genomic loci in HEK293T cells. Editing outcomes are indicated. Data are presented as mean ± SD from $n = 3$ independent biological replicates. Source data are provided as a Source Data file.

resolution of the edited strand flap (Supplementary Fig. 3b). To address this challenge, we hypothesized that inserting a PAM sequence into the edited strand[26,27] might facilitate nicking of the non-edited strand only after resolving the edited strand flap (Supplementary Fig. 3c). Evaluation of VEGFA-3 with new PAM insertion at the targeted strand in HEK293T cells revealed that this type of rPE3b system increased the editing efficiency (Supplementary Fig. 3d).

Moreover, the evopreQ1[7] was used to construct engineered rpegRNA (erpegRNA) to improve rPE (Fig. 2f). The erpegRNAs improved rPE2 editing efficiency by an average of 12-fold in HEK293T cells (Fig. 2g). However, the increased editing efficiency was not observed with the apegRNA strategy[28] (Supplementary Fig. 4a, b). Similarly, erpegRNAs enhanced the editing of rPE3, a result that was further confirmed in HeLa cells (Supplementary Fig. 5a, b). Importantly, erpegRNAs also slightly improved editing purity, particularly at the RP11 locus (Supplementary Fig. 5c, d). Collectively, these results from 16 genomic loci in three cell lines demonstrated the broad applicability of rPE for various types of DNA editing, along with the enhancements provided by ngRNA and erpegRNA in human cells.

## Protein language model-assisted optimization of HNH and MMLV RT for engineered rPE

The rPE employs a Cas9-D10A nickase and RT to edit 3′ terminal DNA of HNH-mediated nick site at the targeted strand. To further optimize rPE functionality, we utilized protein language models (PLMs)[29], which have shown promise in enhancing protein functions, such as the use of uracil-DNA glycosylase for base editing (BE)[30]. Using the Evolutionary Scale Modeling (ESM) framework trained on diverse datasets, including Uniprot and Uniref50, an evolutionary optimization of the HNH and MMLV RT was performed. Three models containing ESM-1v, ESM-MSA-1b, and ESM-IF1[31–33] were utilized for zero-shot variant prediction and inverse folding, applied to all protein variants generated via saturation mutagenesis (Fig. 3a and Supplementary Fig. 6a, b). The top 10% of predicted variants were filtered using a Position-Specific Scoring Matrix, with filtering criteria including higher scores than baseline values and an information value > 0.1 (Fig. 3a–c). Following two rounds of screening, we identified 23 variants with potentially augmented activities of HNH and MMLV RT (Supplementary Data 1). Our results confirmed that the Q826E mutation in HNH led to higher editing efficiency, while the T163E, Q291I, and D339E in MMLV RT also improved efficiency across three genomic loci (Supplementary Fig. 6c, d). We then rationally integrated these four mutations and tested them at the FANCF and HIRA loci. The combination of Q826E, Q291I, and D339E achieved the highest editing efficiency (Fig. 3d, e).

Further optimization was performed based on prior research on variants affecting conformational changes in the HNH domain[10,34]. Among the variants evaluated, only K918A showed an editing efficiency comparable to that of rPE2 (Supplementary Fig. 6e, f). Notably, rPE2max, constructed with optimized codons and the NLS-cMyc from PE2max, exhibited slightly higher efficiency than with the R221K and N394K mutations[4] (Supplementary Fig. 6g, h). We then combined the enhanced variants from PLMs with optimized codons and NLS-cMyc to construct the engineered rPE2max (erPE2max), resulting in a substantial increase in editing efficiency of up to 2.37-fold across ten genomic loci (Fig. 3f, g). Furthermore, we observed an additional

increase of erPE3max editing in HEK293T cells (Fig. 3h). Finally, the compact RTs[10] were also incorporated to enhance the therapeutic potential for rPE system, with evoTf1 exhibiting notably higher editing activity than evoEc48, and a similar efficiency to rPE2 (Supplementary Fig. 7a, b). These findings underscored the enhancement of rPE achieved through PLMs or compact RT across a range of endogenous genomic loci in human cells.

## The rPE with circular rpegRNA and La protein in human cells

The circular PBS and RTT improve the stability of pegRNA, thereby increasing the editing flexibility, but often with similar or decreased efficiency compared to canonical Cas9-directed PE[8]. A recent study documented the enhanced editing ability of circular pegRNA canonical Cas12a-based PE[9]. We hypothesized that the circular RNA structure might facilitate the PBS binding for rPE, thereby improving editing functionality. To test this hypothesis, we constructed the circular erPE2max system, utilizing a split circular rpegRNA in combination with engineered Cas9 and MMLV RT (Supplementary Fig. 8a). Importantly, circular rPE demonstrated a remarkable enhancement of editing efficiency across most genomic loci (Supplementary Fig. 8b). However, the circular PE exhibited decreased efficiency at the FANCF and HIRA loci, in contrast to rPE with the same spacer, which showed increased efficiency (Supplementary Fig. 8c). We additionally developed erPE7max, incorporating improvements from erPE2max along with the addition of the La protein[5]. Use of erPE7max, significantly improved editing efficiency compared to erPE2max, achieving greater enhancement than circular rPE (Fig. 4a, b). La in the middle position could also exert a long linker role to separate Cas9 and the RT domain. Further truncation of MMLV RT in erPE7max slightly increased the editing efficiency (Fig. 4b). Importantly, the increased function of erPE7max compared to rPE7max was also confirmed at five genomic loci (Fig. 4c), and improved activity of erPE7max was further verified at another seven genomic loci (Fig. 4d). Subsequently, we combined the circular RNA with La with the aim of further improving editing efficiency (Supplementary Fig. 8d). Unfortunately, decreased editing efficiency was observed (Supplementary Fig. 8e). In addition, the improvement of erPE7max was also confirmed in HeLa and primary carcinoma-associated fibroblast (CAF) cells (Fig. 4e, f). The addition of ngRNA further enhanced editing efficiency (Fig. 4g). Finally, to further characterize the rPE strategy, we assessed erPE7max at 12 genomic loci without positive selection, with editing efficiencies ranging from 10.08 to 44.41% (Fig. 4h). Collectively, these findings highlight the enhanced editing capabilities of circular erPE2max and erPE7max for rPE.

## Off-target editing analysis for rPE

PE requires complementarity between the target DNA and the spacer of the pegRNA for Cas9 binding, as well as complementarity between the target DNA and the PBS of the pegRNA to initiate pegRNA-templated reverse transcription. Consequently, significantly lower levels of off-target editing have been observed compared to BE and Cas9. However, gRNA-dependent off-target effects may still arise, which are detectable through more precise methods in vitro and in vivo[35,36], and are attributed to the similarity between PBS and spacer sequences. To assess the potential off-target effects of rPE, previously reported off-target loci by PE, including HEK4 and CDH4[36], were tested

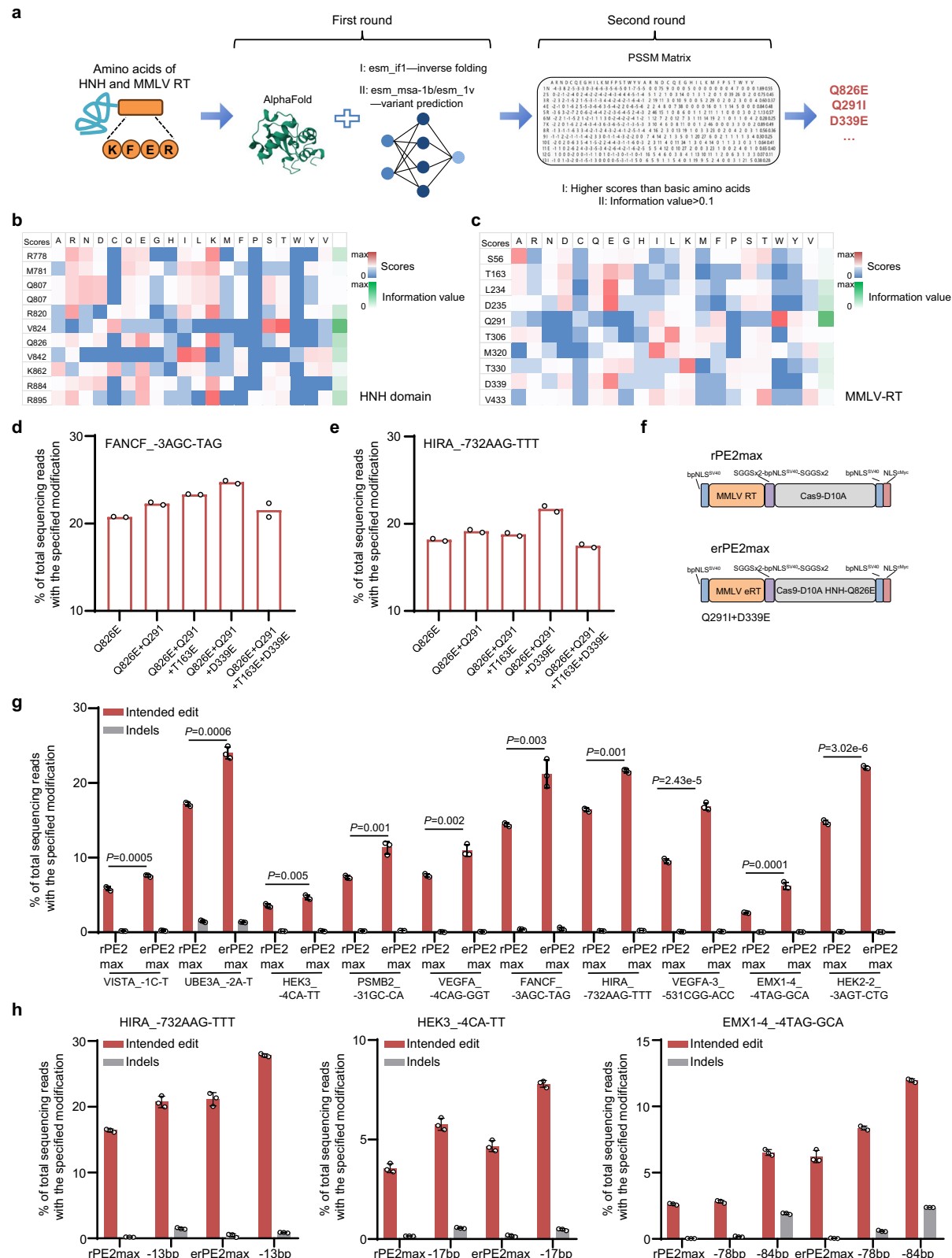

for rPE. Importantly, minimal rPE editing was detected at off-target sites for these two genomic loci with rPE2 and erPE7max, with average editing <1% (Fig. 5a, b). Furthermore, potential off-target sites for two additional genomic loci, FANCF and HIRA, were screened using Cas-OFFinder[37]. Similarly, few instances of off-target editing were detected, even with the enhanced rPE system, with editing efficiency <0.5% at off-

target loci (Fig. 5c, d). More importantly, the specificity between PE and rPE at CDH4 and HEK4 loci with potential off-target effects was assessed, indicating a higher fidelity for rPE compared to PE, especially at the CDH4 locus (Fig. 5e–h). Taken together, these findings suggested that rPE exhibited specific editing with minimal detectable off-target effects.

**Fig. 3 | Protein language model-assisted optimization of HNH and MMLV RT for engineered rPE. a** Schematic of two rounds of PLMs for HNH and MMLV RT evolution. The first round was performed with esm-if1, esm-msa-1b, and esm-1v. The second round was performed with PSSM filtration, with the criteria indicated. Heatmap of PSSM scores for screened amino acids in HNH (**b**) and MMLV RT (**c**) mutants. Red and blue represent higher and lower PSSM scores, respectively, while green represents the information values. Comparison of editing efficiency of rPE across variants at the FANCF (**d**) and HIRA (**e**) loci: Q826E, Q826E + Q291I, Q826E + Q921I + T163E, Q826E + Q921I + D339E, and Q826E + Q921I + T163E + D339E. Editing outcomes are indicated. Data are presented as mean ± SD from $n = 2$ independent biological replicates. **f** Schematic of rPE2 and erPE2max. The blue represents the bpNLS$^{SV40}$, purple represents the SGGSx2-bpNLS$^{SV40}$-SGGSx2 linker,

and red represents the NLS$^{Myc}$. The yellow represents the MMLV RT and its engineered variant (Q291I + D339E), while the gray represents Cas9-D10A and its variant (Q826E). **g** Comparison of editing efficiency (red bars) and indel frequency (gray bars) of rPE2max and erPE2max at 10 genomic loci in HEK293T cells. Editing outcomes are indicated. Data are presented as mean ± SD from $n = 3$ independent biological replicates. Statistical significance was assessed using 2-tailed Student's t-tests. **h** Comparison of editing efficiency (red bars) and indel frequency (gray bars) of rPE2max, erPE2max, and erPE3max at 3 genomic loci in HEK293T cells. Editing outcomes are indicated, and the nick position is specified. Data are presented as mean ± SD from $n = 3$ independent biological replicates. Source data are provided as a Source Data file.

## The rPE installs a functionally enhanced mutation for adoptive cell therapy

To further explore the practical applications and therapeutic potential of rPE, we assessed its efficacy in introducing functionally enhanced variant of *PIK3CD*$^{E527G}$, a mutation that may potentially improve the therapeutic efficacy of adoptive cell therapy[38,39]. Given the substantial improvements observed with circular erPE2max and erPE7max, we introduced the enhanced variant using these mutants, incorporating two base alterations (Fig. 6a). Circular erPE2max and erPE7max successfully introduce the *PIK3CD*$^{E527G}$ with average editing efficiencies of 32.8% and 44.2%, respectively (Fig. 6b). To further validate the utility of the rPE system, we next performed the editing in Jurkat cells using a lentiviral system (Fig. 6c). This system was chosen because of its high transduction efficiency and ability to integrate large DNA sequences into host genomes. Importantly, even after positive selection, the editing efficiency of circular erPE2max reached 31.8%, while erPE7max remained <10% (Fig. 6d). Given the large size of erPE7max, which limited its insertion and expression, we implemented a split strategy at residues 573aa/574aa of Cas9 (Fig. 6c)[35,40]. This design improved the editing efficiency to 40.2%, higher than that achieved by circular erPE2max (Fig. 6d). We then applied the optimized system to primary T cells, adding the spleen focus-forming virus promoter and a green fluorescent protein (GFP) selection marker (Fig. 6e). The enhanced rPE system successfully introduced the PIK3CD variant, achieving an editing efficiency of 40.0% (Fig. 6f, g). More importantly, the functional enhancement of *PIK3CD*$^{E527G}$ was further confirmed through coculture experiments with patient-derived organoids (PDOs), showing a significant increase in apoptotic activity (Fig. 6h, i). Collectively, these results highlighted the ability of rPE to precisely introduce mutations that were associated with enhanced T cell function.

## Discussion

Previous research has indicated that the PAM distal fragment of the R-loop can be released from otherwise stable Cas9:sgRNA:DNA complexes, even after cleavage by RuvC[41]. This fragment may be accessible to prime DNA polymerization during PE editing. However, DNA polymerization in rPE could theoretically be inhibited due to the limited single-strand DNA primer (3 nt) and its proximity to adjacent double-strand DNA. Our findings revealed that rPE exhibited a reverse editing window in human cells, a distinctive feature compared to PE. Notably, rPE strategy at targeted strand might broaden the applicability of PE for other nucleases, such as Ascas12f1[42] and AwaIscB[43], which produces two nicks at the NTS. Additionally, rPE's nickase activity was associated with fewer byproducts compared to PE, potentially enhancing its safety profile for therapeutic applications. It could be attributed to the fact that Cas9-H840A has been reported to induce unwanted DSBs in PE[23], further explaining the poor reverse editing seen with PE2 and rpegRNA. However, rPE's editing efficiency is generally lower than that of PE at proximal editing sites, likely due to limitations in its polymerization process. This restriction underscores the need for further optimization through molecular mechanism analysis.

The increased editing efficiency with the deletion of RNaseH in MMLV RT for rPE2 was not observed[35,40,44], likely due to spatial hindrance caused by the close proximity between Cas9 and the catalytic domain of the truncated RT, which may impair its functionality. As expected, leveraging the rPE3 and erpegRNA strategies improved rPE function and specificity. More importantly, the rPE3b system was implemented by inserting a new PAM sequence, not considering that the edit must lie within a second protospacer. Previous studies have indicated that introducing additional same-sense variants in RTT can prevent the reversal of PE by MMR[4,28]. Accordingly, the rPE3b strategy is promising for achieving higher editing efficiency in rPE target with potential same-sense variants. Regrettably, the apegRNA strategy in the rPE system did not yield satisfactory results, likely due to structural alterations in the arpegRNA, which may hinder its binding to distal sequences.

Importantly, several unreported variants within HNH nuclease and MMLV RT were identified, which significantly improved rPE efficiency, highlighting the potential of PLM to optimize gene editors. However, the combination of all enhanced mutations did not lead to a further increase in editing efficiency, suggesting that a more rational design based on structural and functional insights is necessary for protein optimization. The integration of circular RNA or La with rPE significantly enhanced editing efficiency. The circular rPE structure improved efficiency at most loci, though the corresponding circular PE exhibited a notable decrease in efficiency. We predict that the circular rPE may stabilize the rpegRNA and facilitate DNA polymerization, thus enhancing rPE editing efficiency. The differences in nickase activity between Cas9-H840A and Cas9-D10A may also contribute to the observed discrepancy in efficiency.

We further evaluated reported and screened off-target loci to assess off-target effects. Importantly, minimal off-target editing across these genomic loci and a potentially higher fidelity was observed compared to PE, underscoring the safety profile of rPE. Finally, the enhanced rPE system demonstrated therapeutic potential by efficiently inserting a functional variant relevant to adoptive cell therapy. Notably, we chose the lentiviral system for delivery due to the potentially lower efficiency of long rPE mRNA and rpegRNA synthesis, as well as the limited availability of electroporation platforms in many laboratories. When delivered via the lentiviral system, the split circular rPE2max achieved editing efficiencies of up to 30% in Jurkat cells. While its efficiency was lower than the split erPE7max, which reached up to 40% in primary T cells, the finding underscored the promising application potential of circular rPE. Future efforts could focus on optimizing the formation and delivery of circular rpegRNA to further enhance its editing efficiency. In particular, the split erPE7max exhibited robust editing performance in T cells. Delivering this system as an electroporated RNP complex, especially when combined with poly(U) tails, might further improve its efficiency[5]. This split and tightly regulated system might hold promise for advancing immune cell engineering through high-throughput screening[45].

In summary, our study introduced an innovative variant of PE, rPE, showcasing its substantial editing efficiency with a reverse editing window and its practical applicability. The conceptual design behind rPE also opened new possibilities for future modifications of PE tools, further expanding and enriching the gene editing toolset.

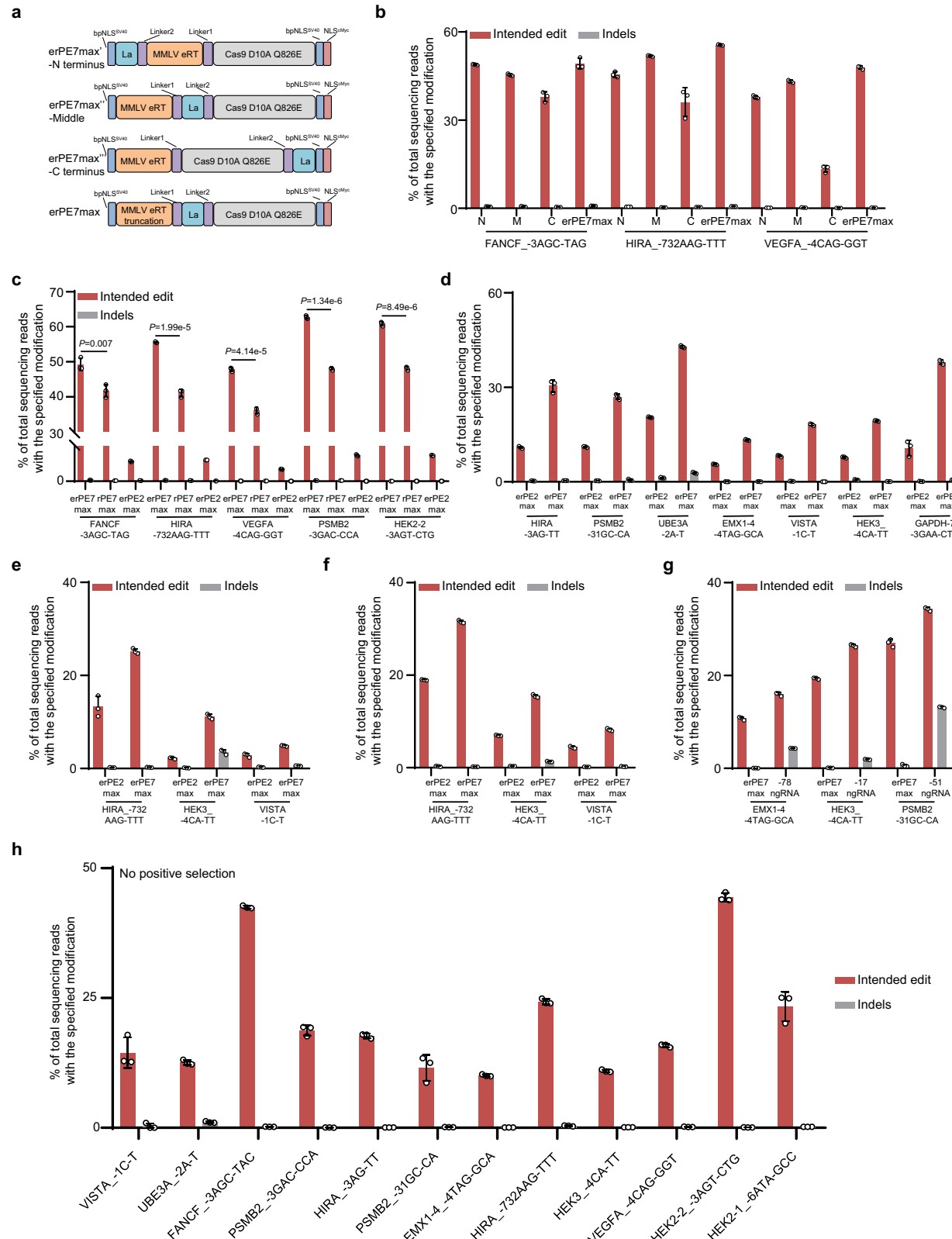

## Methods

### Ethics statement

The study complied with all relevant ethical regulations for research involving human participants. Ethical approval was obtained from the Ethics Committee of Tianjin Medical University Cancer Institute and Hospital, and written informed consent was obtained from all participants. Sex of human samples in this study, including PDAC tissues and PBMCs, was not considered as a variable, as these in vitro models are not influenced by sex-specific biological factors.

### Cell culture and transfection

HEK293T (ATCC® CRL-3216™), HeLa (ATCC® CCL-2™), Jurkat (ATCC® TIB-152™) and HepG2 (ATCC® HB-8065™) cell lines were obtained from the American Type Culture Collection (ATCC, Manassas, VA, USA).

**Fig. 4 | The rPE with circular rpegRNA and La protein in human cells.**
**a** Schematic of erPE7max variants with different arrangements. Dark blue represents bpNLSSV40, purple represents the linkers, yellow represents MMLV eRT (Q291I + D339E) and its truncated form, gray represents the Cas9-D10A variant (Q826E), and light blue represents La protein. **b** Comparison of editing efficiency (red bars) and indel frequency (gray bars) of erPE7max variants at 3 genomic loci in HEK293T cells. Editing outcomes are indicated. Data are presented as mean ± SD from $n = 3$ independent biological replicates. **c** Comparison of editing efficiency (red bars) and indel frequency (gray bars) between erPE7max and rPE7max at 5 genomic loci in HEK293T cells. Editing outcomes are indicated. Data are presented as mean ± SD from $n = 3$ independent biological replicates. Statistical significance was assessed using 2-tailed Student's t-tests. **d** Comparison of editing efficiency (red bars) and indel frequency (gray bars) between erPE7max and erPE2max at 7 genomic loci in HEK293T cells. Data are presented as mean ± SD from $n = 3$ independent biological replicates. Editing efficiency (red bars) and indel frequency (gray bars) between erPE2max and erPE7max at 3 genomic loci in HeLa (**e**) and primary CAF (**f**) cells. Data are presented as mean ± SD from $n = 3$ independent biological replicates. **g** Editing efficiency (red bars) and indel frequency (gray bars) between erPE7max and the addition of ngRNA at 3 genomic loci in HKE293T cells. Data are presented as mean ± SD from $n = 3$ independent biological replicates. **h** Editing efficiency (red bars) and indel frequency (gray bars) with erPE7max across 12 genomic loci in HEK293T cells. Data are presented as mean ± SD from $n = 3$ independent biological replicates. Source data are provided as a Source Data file.

HEK293T, HeLa and HepG2 cells were cultured in DMEM, while Jurkat was cultured with RPMI1640 supplemented with 10% FBS in the humidified incubator equilibrated with 5% $CO_2$ at 37 °C. For transfected experiments, cells were seeded in 24-well plates (Corning, USA) and performed using lipofectamine 3000 (Thermo Scientific, USA) based on the manufacturer's instructions. Briefly, 600 ng of PE or rPE editor and 300 ng of pegRNA or rpegRNA -expressing plasmid were together transfected with 50 μl of Opti-MEM (Gibco, USA) containing lipo 3000 and P3000 for 24 h. 100 ng ngRNA will be added with rPE3 system. The same amount of split rPE and circular RNA was used for circular rPE and PE system. After transfection, cells were cultured in fresh medium containing 5 μg/ml puromycin (Merck, USA) for 3–4 days. This selection step was used to increase the sensitivity of early-stage experiments by enriching for edited cells, ensuring more reliable detection of editing events. Additionally, selection-free experiments were conducted to assess editing efficiency under standard, unselected conditions (indicated in figure legend). Finally, genomic DNA was extracted via QuickExtract DNA Extraction Solution (Epicentre, USA). On-target genomic regions (200 bp–300 bp) of interest were amplified by PCR for high-throughput DNA sequencing.

### Plasmid construction
Compact reverse transcriptase and multiple mutations part for rPE system were synthesized by AZENTA. PCR products were gel purified, digested with DpnI restriction enzyme (NEB, USA), and assembled via Gibson or Goldengate assembly based on the manufacturer's instructions. All gRNA-expression plasmids were assembled via Golden Gate with the protospacer sequence embedded in the primers, and RNF2 sgRNA expression plasmids were used as the template[25]. The main primers are listed in Supplementary Data 2.

### Strains and culture conditions
E. coli Trans5α was used as the cloning host and cultured at 37 °C in lysogeny broth (LB, 1% (w/v) tryptone, 0.5% (w/v) yeast extract, and 1% (w/v) NaCl). 100 mg/L ampicillin (Sigma, USA) was used for screen of positive cloning.

### High-throughput DNA sequencing of genomic DNA samples and data analysis
Next-generation sequencing library preparations and analysis were performed as previously reported[46]. Briefly, purified PCR fragments were treated in one reaction with End Prep EnzymeMix for end repair, 5′ phosphorylation and dA tailing, which was followed by T-A ligation to add adapters to both ends, of which PCR products were purified and quantified. Then the sequencing was carried out on Illumina HiSeq instrument according to the manufacturer's instructions.

Analysis of amplicon sequencing data was performed using CRISPResso2 v.2.0.45 in batch mode[47] and analyzed[2]. For all PE yield quantification, PE efficiency was calculated as: percentage of (number of reads with the desired edit that do not contain indels)/(number of total reads). For all experiments, indel yields were calculated as: (number of indel containing reads)/(total reads). All genomic loci and deep sequencing oligos of pegRNA or rpegRNA are listed in Supplementary Data 3.

### Evaluation of potentially contiguous SSBs for PE and rPE
An indirect method was utilized to evaluate the contiguous SSBs for PE and rPE. Firstly, ~500 bp DNA sequence (50 bp genome sequence and 450 bp vector sequence) including the intended PE or rPE editing outcome was inserted into HEK293T genome using lentivirus. PE editing outcome was designed without editing of PAM or its adjacent bases, while the rPE editing was selected within the spacer sequence. Subsequently, the modified HEK293T cells were transfected with indicated PE/rPE system and another ngRNA for 5 days. The indel frequency was determined using high-throughput DNA sequencing, serving as an indirect indicator to evaluate the contiguous SSBs of PE/rPE.

### Selection of genomic loci with differential transcriptional status
Firstly, the promoter of gene *GAPDH* and *FOXA1* were selected considering their expression level in HEK293T cells. Then the differential transcriptional status was confirmed with H3K27ac ChIP-seq data from UCSC in hg38 human genome. The TSSs (transcription start sites) were determined utilizing the marked sequence from NCBI.

### PLMs evolution for HNH and MMLV RT
The evolutionary screening process mainly consists of two rounds. In the first round, we evaluate the protein structure and fitness landscape to screen potential efficient mutation sites. Alphafold2 is used to generate the overall structure of enzymes[48], determining the binding sites of enzymes with substrates and the amino acid sites in disordered regions. Subsequently, we use three models, ESM-1v, ESM-MSA-1b and transformer-inverse fold (ESM-if1), to evaluate the functional scores of given amino acid sequences with saturation mutations. We combine the results from the first round to comprehensively select a top 10% mutation sites. In the second round, we filter through PSSM scoring matrix (https://possum.erc.monash.edu/index.jsp) considering the higher score compared to existing amnio acid to obtain potential efficient mutation sites for HNH and MMLV RT. The top variants of MMLV were further rationally selected based on its functional domain. The screened data of PLMs are listed in the Supplementary Data 1.

### Off-target analysis
HEK4-2 and CDH4 potential off-target loci were obtained from the ref. 36. Off-target loci of FANCF and HIRA were analyzed by Cas-OFFinder[37] where potential off-target loci containing the most similar sequences of selected target loci were chosen as predicted off-target sites. The off-target effects were evaluated through the indel frequency and intended nucleotide conversion at tested off-target loci[5]. Briefly, reads were aligned to corresponding off-target reference sequences and each off-target amplicon sequence was compared with the 3′ DNA flap sequence encoded by the rpegRNA/pegRNA extension. Any reads with this nucleotide converted to that on the 3′ DNA flap were considered off-target, which were calculated from the output file

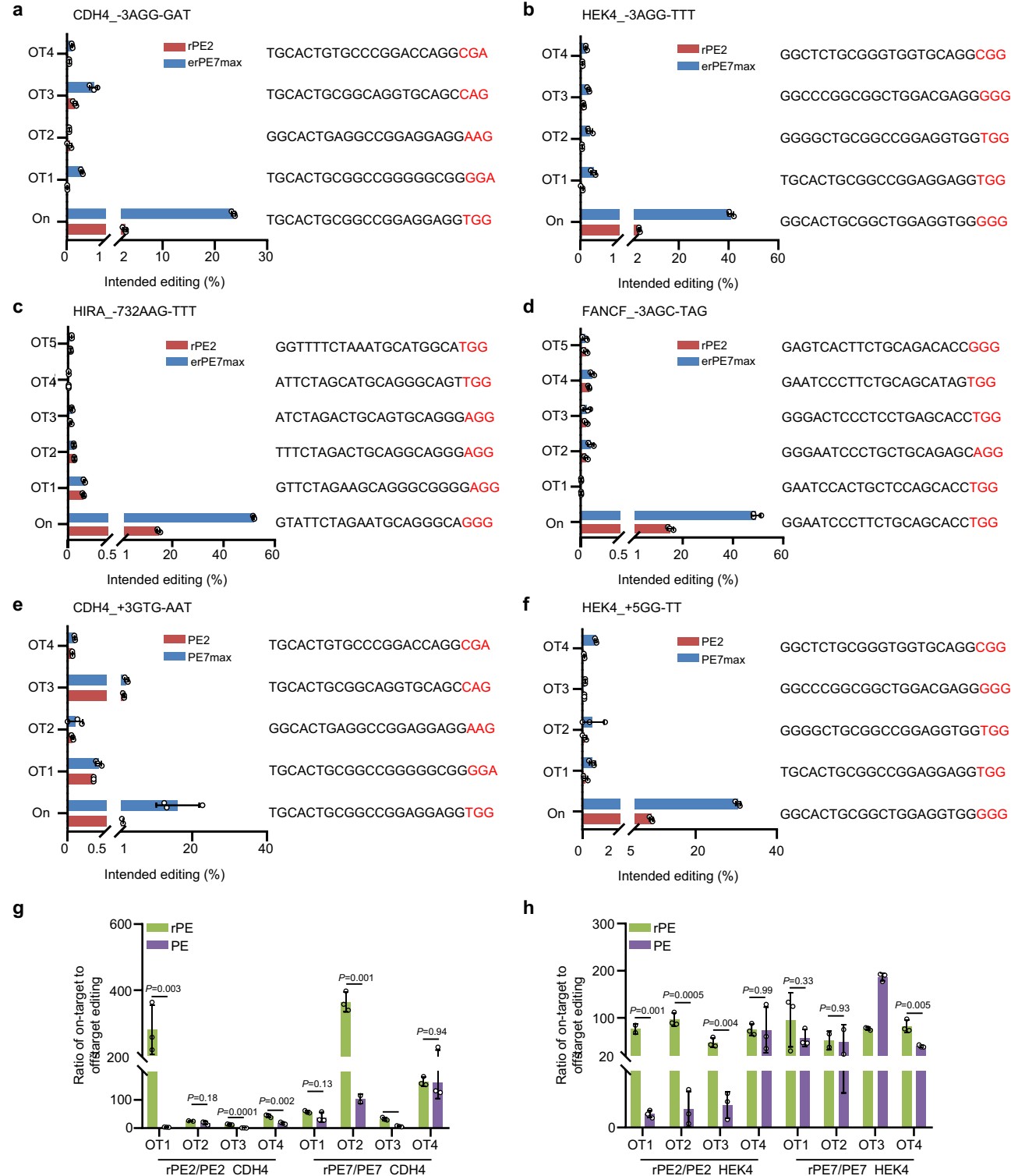

**Fig. 5 | Off-target editing analysis for rPE. a** Editing efficiency of rPE2 (red bars) and erPE7max (blue bars) for on-target and off-target sites at CDH4 (**a**) and HEK4 (**b**) genomic loci in HEK293T cells. The off-target sequences are indicated on the right. Data are presented as mean ± SD from *n* = 3 independent biological replicates. Editing efficiency of rPE2 (red bars) and erPE7max (blue bars) for on-target and off-target sites at HIRA (**c**) and FANCF (**d**) genomic loci in HEK293T cells. The off-target sequences are indicated on the right. Data are presented as mean ± SD from *n* = 3 independent biological replicates. Editing efficiency of PE2 (red bars) and PE7max (blue bars) for on-target and off-target sites at CDH4 (**e**) and HEK4 (**f**) genomic loci in HEK293T cells. The off-target sequences are indicated on the right. Data are presented as mean ± SD from *n* = 3 independent biological replicates. Off-target specificity between the PE and rPE systems at CDH4 (**g**) and HEK4 (**h**) genomic loci in HEK293T cells. Data are presented as mean ± SD from *n* = 3 independent biological replicates. Statistical significance was assessed using 2-tailed Student's t-tests. Source data are provided as a Source Data file.

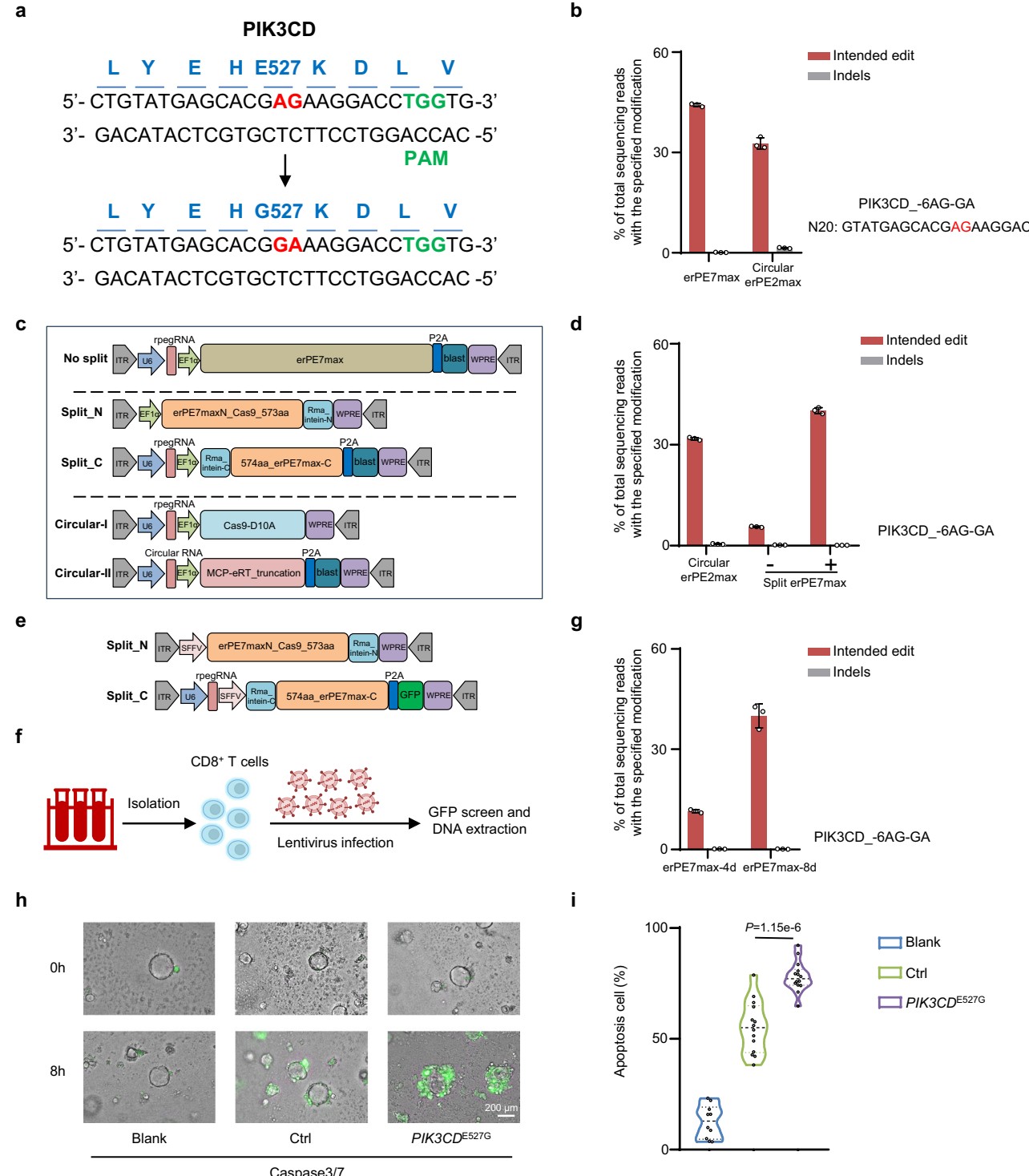

**a** PIK3CD

**b** PIK3CD_-6AG-GA
N20: GTATGAGCACGAGAAGGACC

**c** 

**d** PIK3CD_-6AG-GA

**e** 

**f** CD8⁺ T cells

**g** PIK3CD_-6AG-GA

**h** Caspase3/7

**i** *P*=1.15e-6

"Nucleotide_frequency_summary_around_sgRNA". These off-target sites and the associated primers are listed in the Supplementary Data 4.

### Lentivirus infection and prime editing of PIK3CD in Jurkat and primary T cells

The generation of lentiviruses was conducted according to the previous reports[46]. Briefly, constructed vectors containing rPE system and indicated rpegRNA together with psPAX2 and pMD2.G, were co-transfected into the packaging cell line HEK293T at a weight ratio of 3:2:1. Viral supernatants were collected 48 h later, clarified by filtration, and concentrated by ultracentrifugation. Then the concentrated

viruses were used to infect $5 \times 10^5$ Jurkat cells (20–30% confluence) in a 60-mm dish with 5 mg/mL polybrene using MOI = 20. Infected Jurkat cells were selected by 2 μg/ml blasticidin (Solarbio, China) to the culture medium and then cultured for another 5 days. The cells were collected and the genomic DNA was subjected to deep sequencing to measure the editing efficiency.

Human CD8⁺ T cells were purified from healthy donors and cultured as previously reported[49]. Breifly, PBMCs were isolated by density gradient centrifugation over Ficoll Paque. And then CD8⁺ T cells were isolated by CD8⁺ T Cell Isolation Kit (Miltenyi Biotec, Germany) under sterile conditions following the instructions of the manufacturer.

**Fig. 6 | The rPE installs a functionally enhanced mutation for adoptive cell therapy. a** Schematic of the PIK3CD_E527G mutation induced by the rPE strategy. The mutated amino acids are marked in blue, the edited bases in red, and the PAM sequence in green. **b** Comparison of editing efficiency (red bars) and indel frequency (gray bars) between erPE7max and circular erPE2max at the PIK3CD locus in HEK293T cells. Editing outcomes are indicated. Data are presented as mean ± SD from *n* = 3 independent biological replicates. **c** Schematic of no-split erPE7max (upper), split erPE7max (middle), and circular split erPE2max (lower) systems for lentivirus production and Jurkat cell transduction. Elements of the lentivirus components are marked in different colors. **d** Comparison of editing efficiency (red bars) and indel frequency (gray bars) across circular erPE2max, no-split, and split erPE7max systems at the PIK3CD locus in Jurkat cells. Editing outcomes are indicated. Data are presented as mean ± SD from *n* = 3 independent biological replicates. **e** Schematic of the split erPE7max (middle) system for lentivirus production and T cell transduction. Elements of the lentivirus components are marked in different colors. **f** Schematic of primary CD8 + T cell isolation, transduction, and sequencing. **g** Editing efficiency (red bars) and indel frequency (gray bars) with erPE7max (4 d and 8 d) at the PIK3CD locus in primary CD8 + T cells. Editing outcomes are indicated. Data are presented as mean ± SD from *n* = 3 independent biological replicates. **h** Phenotypic outcomes of PDOs cocultured with CD8 + T cells with and without rPE editing at 0 and 8 h. Blue represents Caspase3/7 activity. **i** Apoptosis cell percentage across blank, ctrl, and PIK3CD_E527G groups. The dot represents the apoptosis cell percentage from a single PDO. Data are presented as means ± SD (n = 10 for blank, *n* = 14 for ctrl, *n* = 14 for PIK3CD_E527G). Statistical significance was assessed using 2-tailed Student's t-tests. Source data are provided as a Source Data file.

The CD8⁺ T cells were cultured in T cell-specific medium and transduced with lentivirus using MOI = 20 after 2 days activation by CD3/CD28 Dynabeads (Thermo Scientific, USA). Subsequently, the CD8⁺ T cells were further sorted based on GFP signaling after an 8-day culture, and genome DNA was extracted by Genomic DNA Mini Kit (Thermo Scientific, USA) and subjected to deep sequencing.

### Lentivirus infection and prime editing in primary CAFs

Tissues underwent digestion using a mixture of 1 mg/mL collagenase I (Sigma, USA) and 1.5 mg/mL hyaluronidase (Sigma, USA) in DMEM supplemented with 10% fetal bovine serum at 37 °C with agitation for 3 h. Subsequently, the supernatant containing stromal cells was collected and centrifuged at 250 × *g* for 5 min. After discarding the supernatant, the pellet was resuspended and cultured in 10% FBS-DMEM at 37 °C with 5% CO2. This method facilitated the successful isolation of primary CAFs from the tumor microenvironment, allowing subsequent in vitro investigations. Then indicated lentivirus containing the rPE and rpegRNA were used to infect about 5 × 10⁵ cells with polybrene. Infected cells were selected by 2 μg/ml blasticidin (Solarbio, China) to the culture medium and then cultured for another 4 days. The cells were collected and the genomic DNA was subjected to deep sequencing to measure the editing efficiency.

### The establishment and culture of patient-derived organoids (PDOs)

The PDOs were established and cultured as previously reported[50]. Briefly, fresh PDAC specimens were washed three times with cold PBS containing 10% penicillin and streptomycin. The specimens were separated into small pieces (<1mm³) with sterile blades, and then digested with a mixture of 5 mg/mL collagenase II, 5 mg/mL collagenase IV and 5 mg/mL collagenase XI for 30 min at 37 °C under severe vibration. Advanced DMEM/F12 (Gibco, USA) containing 7.5% BSA was added to terminate digestion. Tumor cells were harvested, 400 RCF, 5 min. Subsequently, tumor cells were embedded in Matrigel (Corning, USA) and grown in Human Complete Feeding Medium (HCPLT). Depending on growth, the organoid medium was changed approximately every 2 days, and organoids were passaged approximately every 14 to 20 days.

### Organoids apoptosis assay

For assaying the apoptosis of organoids, organoids were isolated by TrypLE Express (Gibco, USA) and plated at 1 × 10⁴/ml on 60% Matrigel-coated black clear bottom 96-well plates (Perkin Elmer, USA). After 48 h, the organoids were pelleted and stabilized, then CD8⁺ T cells from different groups were added with a ratio 10:1. After incubation for 8 h, the cells were washed and treated with CellEvent™Caspase-3/7 Green ReadyProbes (Invitrogen, USA) for 45 min. The apoptosis of organoids was monitored by BZ-X800 fluorescence microscope. The probe had a maximum excitation wavelength of 502 nm and a maximum emission wavelength of 530 nm.

### Statistics and reproducibility

Unless otherwise noted, all data are presented as means ± s.d. and analyzed with statistical methods from three independent experiments. The significance of the difference between the control and experiment group was calculated via student's *t* test using GraphPad Prism 8. *P* < 0.05 was considered to be statistically significant. No data was excluded from the analyses. Samples were not randomized. Investigators were not blinded during experiments and data analysis.

### Reporting summary

Further information on research design is available in the Nature Portfolio Reporting Summary linked to this article.

## Data availability

There is no restriction on the data associated with this study. HTS data generated in this study have been deposited in the NCBI Sequence Read Archive database under accession code PRJNA1099390. Source data are provided with this paper.

## Code availability

The source code for AlphaFold2 is available at the GitHub repository https://github.com/google-deepmind/alphafold under the Apache 2.0 license and has been archived at Zenodo with the https://doi.org/10.5281/zenodo.6998041. The source code for ESM-MSA-1b, ESM-1v and ESM-IF1 is available at the GitHub repository https://github.com/facebookresearch/esm and has been archived at Zenodo with the https://doi.org/10.5281/zenodo.7566740. In accordance with license terms, original copyright and license statements were retained within all reused source files. Attribution has been provided in compliance with the respective open-source licenses.

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

## Acknowledgements

This work was supported by the National Key Research and Development Program of China (2021YFA1201100 to J.H.), the National Natural Science Foundation of China (grants 32471478 to C.Y., 82203238 to C.Y., 82430057 to J.H., 82273362 to X.W., 82272680 to C.H. and 82271895 to Y.F.), Tianjin Natural Science Foundation (No.24JCQNJC00390 to T.Z.) and Tianjin Key Medical Discipline-Specialty Construction Project (TJYXZDXK-OO9A). The PLM in this work was supported by High-performance Computing Platform of Tianjin Medical University. We thank International Science Editing (http://www.internationalscienceediting.com) for editing this manuscript.

## Author contributions

C.Y., L.S., C.B., X.Z., J.Y., and J.H. designed the research and drafted the manuscript. C.Y., Q.F., M.L., J.Z., L.R., and T.Z. performed the experiments studies, the computational studies, and analyzed data. K.W., J.D., X.W., C.H., Y.F., and Z.X. guided the research. All authors agreed the final version of this manuscript.

## Competing interests

C.Y., J.Y., and J.H. have submitted a patent application to the China National Intellectual Property Administration (CNIPA) pertaining to the optimization of prime editing in the context of gene editing (application number ZL 2024 1 0172211.7). The remaining authors declare no competing interests.
