## [Peer Review file · Nature Communications]

Prime Editor with Rational Design and AI-Driven Optimization for Reverse Editing Window and Enhanced Fidelity

Corresponding Author: Professor Changhao Bi

Version 1:

Reviewer comments:

Reviewer #4

(Remarks to the Author)

Comment 1:

“Accordingly, the efficient editing scope of canonical PEs could be further expanded, and the potential SSBs might be mitigated to further improve the safety profile for therapeutic employment”

The following statement should not be included, as the paper does not address the potential SSBs that is produced by canonical PEs. As reviewer 3 mentioned, canonical PEs can be used to disrupt the PAM sequence to prevent SSBs. rPE is also very similar in its ability to produce SSBs in the DNA. This sentence insinuates that this is a property that the authors address in this work.

“Notably, rPE’s PBS designed based on the 5’ direction DNA sequence of HNH mediated nick site, differs from the spacer sequence, which in future studies is expected to further reduce gRNA-dependent off-target effects.”

This does not make sense. Why would the PBS sequence differing from the spacer sequence reduce gRNA dependent off-target effects? The PBS still binds to a region of DNA in the genome in rPE just like canonical PE if this is what the authors are referring to. This statement should not be included, especially since there is no evidence to back this up.

Comment 2:

To make any comments about active transcriptional status and editing efficiencies, analysis needs to be done at more sites (not just two) followed by statistical analysis for significance. PE is a dynamic process with editing efficiencies varying by many factors including PBS, RTT, DNA repair, etc. The authors could expand this analysis to other sites or change the text to say that they performed it for a few sites, the trends show that there may be correlation, but more sites need to be tested to make any conclusive statement.

Comment 3:

“The increased fidelity could be attributed to the fact that Cas9-H840A has been reported to induce unwanted DSBs in PE, further explaining the poor reverse editing seen with PE2 and rpegRNA. ”

This statement does not accurately reflect the authors' intended message. It is sufficient to mention that the observed editing with H840A-nickase-based editors likely results from the known capability of the H840A nickase to introduce double-strand breaks (DSBs). Introducing the concept of increased fidelity in this context is confusing and should be avoided.

Second Revision:

1) I agree that puromycin selection can be standard in the field and is totally okay to use for experiments as long as it is clearly indicated that this was done in the methods section.

2) The authors should remove all comparisons between erPE7 and PE7. Simply stating that erPE7 expands the editing potential of prime editors is sufficient to highlight the utility of the method. Reviewer 3 raised a crucial point regarding pegRNA design. Not only were the pegRNAs incorrectly designed—a significant oversight—but comparing differently designed pegRNAs with two distinct editors does not constitute a fair ("apples-to-apples") comparison. Given that editing efficiencies in prime editing vary widely depending on pegRNA design, it is likely that additional screening could identify pegRNAs for PE7 that outperform erPE7. The authors likely designed an optimal pegRNA for the erPE7 strategy while selecting a suboptimal pegRNA for PE7. While commenting on indel frequencies is acceptable since the same spacer is used, comparisons of editing efficiencies between editors requiring different pegRNAs should not be included. For the manuscript to be seriously considered for acceptance, all such comparisons between erPE7 and PE7 must be completely removed.

Version 2:

Reviewer comments:

Reviewer #4

(Remarks to the Author)

The authors have addressed my concerns.

Authors: We sincerely appreciate Reviewer #4's constructive feedback. We have carefully addressed each comment below, and all corresponding changes have been incorporated in the revised manuscript. For clarity, new or modified texts in the main manuscript are highlighted in blue.

Reviewers' comments:

Reviewer #4 (Remarks to the Author):

Comment 1:

“Accordingly, the efficient editing scope of canonical PEs could be further expanded, and the potential SSBs might be mitigated to further improve the safety profile for therapeutic employment”

The following statement should not be included, as the paper does not address the potential SSBs that is produced by canonical PEs. As reviewer 3 mentioned, canonical PEs can be used to disrupt the PAM sequence to prevent SSBs. rPE is also very similar in its ability to produce SSBs in the DNA. This sentence insinuates that this is a property that the authors address in this work.

Author: Thanks for the reviewer's suggestion. The sentence was removed in the revised manuscript to avoid overstatement.

“Notably, rPE's PBS designed based on the 5' direction DNA sequence of HNH mediated nick site, differs from the spacer sequence, which in future studies is expected to further reduce gRNA-dependent off-target effects.”

This does not make sense. Why would the PBS sequence differing from the spacer sequence reduce gRNA dependent off-target effects? The PBS still binds to a region of DNA in the genome in rPE just like canonical PE if this is what the authors are referring to. This statement should not be included, especially since there is no evidence to back

this up.

Author: Thanks for the reviewer’s suggestion. To ensure scientific rigor and clarity, the sentence has been removed from the revised manuscript.

Comment 2:

To make any comments about active transcriptional status and editing efficiencies, analysis needs to be done at more sites (not just two) followed by statistical analysis for significance. PE is a dynamic process with editing efficiencies varying by many factors including PBS, RTT, DNA repair, etc. The authors could expand this analysis to other sites or change the text to say that they performed it for a few sites, the trends show that there may be correlation, but more sites need to be tested to make any conclusive statement.

Author: Thanks for the reviewer’s suggestion. To strengthen the robustness of our conclusions, we have expanded our analysis to include 12 different genomic sites. Editing efficiencies were quantified and subjected to statistical analysis. However, no significant difference of editing efficiency was observed across genomic loci with differential transcriptional status (**Supplementary figure 1j-k**), suggesting a more complex underlying mechanism.

Comment 3:

“The increased fidelity could be attributed to the fact that Cas9-H840A has been reported to induce unwanted DSBs in PE, further explaining the poor reverse editing seen with PE2 and rpegRNA. ”

This statement does not accurately reflect the authors' intended message. It is sufficient to mention that the observed editing with H840A-nickase-based editors likely results from the known capability of the H840A nickase to introduce double-strand breaks (DSBs). Introducing the concept of increased fidelity in this context is confusing and should be avoided.

Author: Thanks for the reviewer's suggestion. The phrase "increased fidelity" has been removed to prevent misinterpretation. The revised sentence reads: "It could be attributed to the fact that Cas9-H840A has been reported to induce unwanted DSBs in PE, further explaining the poor reverse editing seen with PE2 and rpegRNA."

Second Revision:

1) I agree that puromycin selection can be standard in the field and is totally okay to use for experiments as long as it is clearly indicated that this was done in the methods section.

Author: We appreciate the reviewer's acknowledgment. The use of puromycin selection has been clearly described in the revised Methods section.

2) The authors should remove all comparisons between erPE7 and PE7. Simply stating that erPE7 expands the editing potential of prime editors is sufficient to highlight the utility of the method. Reviewer 3 raised a crucial point regarding pegRNA design. Not only were the pegRNAs incorrectly designed—a significant oversight—but comparing differently designed pegRNAs with two distinct editors does not constitute a fair ("apples-to-apples") comparison. Given that editing efficiencies in prime editing vary widely depending on pegRNA design, it is likely that additional screening could identify pegRNAs for PE7 that outperform erPE7. The authors likely designed an optimal pegRNA for the erPE7 strategy while selecting a suboptimal pegRNA for PE7. While commenting on indel frequencies is acceptable since the same spacer is used, comparisons of editing efficiencies between editors requiring different pegRNAs should not be included. For the manuscript to be seriously considered for acceptance,

all such comparisons between erPE7 and PE7 must be completely removed.

Author: Thank you for the reviewer's valuable comments. As suggested, all direct comparisons between erPE7 and PE7, including those in the Abstract, Results, and Discussion sections, have been removed. We now focus solely on the unique editing capabilities of erPE7 without drawing direct comparisons that could be confounded by pegRNA/rpegRNA design variability. We appreciate the reviewer for pointing out this critical issue, which has helped improve the fairness and clarity of our data interpretation.

We are grateful for the reviewer's insightful and constructive feedback, which has significantly improved the quality and rigor of our manuscript. We hope that the revised version sufficiently addresses all concerns and that our work will now be suitable for publication in *Nature communications*.